# Comprehensive multimodality characterization of hemodynamically significant and non-significant coronary lesions using invasive and noninvasive measures

Leif-Christopher Engel[1,2,3]*, Ulf Landmesser[1,2], Youssef S. Abdelwahed[1], Milosz Jaguszewski[1,4], Kevin Gigengack[1], Thomas-Heinrich Wurster[1], Carsten Skurk[1], Costantina Manes[1], Andreas Schuster[5,6], Michel Noutsias[7], Bernd Hamm[8], Rene M. Botnar[9,10], Marcus R. Makowski[8], Boris Bigalke[1]

1 Charité Campus Benjamin Franklin, Universitätsmedizin Berlin, Klinik für Kardiologie, Berlin, Germany, 2 Berlin Institute of Health (BIH), Berlin, Germany, 3 Klinik für kardiovaskuläre Erkrankungen, Deutsches Herzzentrum München (DHM), Germany, 4 Medical University of Gdansk, Gdańsk, Poland, 5 Department of Cardiology and Pulmonology, Georg-August-University, Göttingen, Germany, 6 Department of Cardiology, Royal North Shore Hospital, The Kolling Institute, Northern Clinical School, University of Sydney, 5th Floor, Acute Services Building, Reserve Road, St Leonard's, Sydney, Australia, 7 Mid-German Heart Center, Department of Internal Medicine III (KIM-III), Division of Cardiology, Angiology and Intensive Medical Care, University Hospital Halle-Wittenberg, Mid-German Heart Center, Martin-Luther-University Halle-Wittenberg, Halle (Saale), Germany, 8 Charité Campus Benjamin Franklin, Universitätsmedizin Berlin, Klinik für Radiologie, Berlin, 9 King's College London, Division of Imaging Sciences and Biomedical Engineering, London, England, United Kingdom, 10 Pontificia Universidad Católica de Chile, Escuela de Ingeniería, Santiago, Chile, Germany

☯ These authors contributed equally to this work.
* leifengel@hotmail.com

**Data Availability Statement:** Data cannot be shared publicly because of european data

## Abstract

### Background

There is limited knowledge about morphological molecular-imaging-derived parameters to further characterize hemodynamically relevant coronary lesions.

### Objective

The aim of this study was to describe and differentiate specific parameters between hemodynamically significant and non-significant coronary lesions using various invasive and non-invasive measures.

### Methods

This clinical study analyzed patients with symptoms suggestive of coronary artery disease (CAD) who underwent native T1-weighted CMR and gadofosveset-enhanced CMR as well as invasive coronary angiography. OCT of the culprit vessel to determine the plaque type was performed in a subset of patients. Functional relevance of all lesions was examined using quantitative flow reserve (QFR-angiography). Hemodynamically significant lesions

protection guidelines. Data are stored on encrypted hard drives and safely locked up in special storage rooms at Charite Hospital only available for researchers who meet the criteria for access to confidential data. However interested researchers could contact the local ethics committee in order to make data access requests. Data access request may be made via email (ethikkommission@charite. de or manuela.pueschel@charite.de).

**Funding:** The authors received no specific funding for this work.

**Competing interests:** The authors have declared that no competing interests exist.

**Abbreviations:** CAD, coronary artery disease; CK, creatine kinase; ACEI, angiotensin converting enzyme inhibitor; ARB, angiotensin receptor blocker.

were defined as lesions with a QFR <0.8. Signal intensity (contrast-to-noise ratios; CNRs) on native T1-weighted CMR and gadofosveset-enhanced CMR was defined as a measure for intraplaque hemorrhage and endothelial permeability, respectively.

## Results

Overall 29 coronary segments from 14 patients were examined. Segments containing lesions with a QFR <0.8 (n = 9) were associated with significantly higher signal enhancement on Gadofosveset-enhanced CMR as compared to segments containing a lesions without significant stenosis (lesion-QFR>0.8; n = 19) (5.32 (4.47–7.02) vs. 2.42 (1.04–5.11); p = 0.042). No differences in signal enhancement were seen on native T1-weighted CMR (2.2 (0.68–6.75) vs. 2.09 (0.91–6.57), p = 0.412). 66.7% (4 out of 6) of all vulnerable plaque and 33.3% (2 out of 6) of all non-vulnerable plaque (fibroatheroma) as assessed by OCT were hemodynamically significant lesions.

## Conclusion

The findings of this pilot study suggest that signal enhancement on albumin-binding probe-enhanced CMR but not on T1-weighted CMR is associated with hemodynamically relevant coronary lesions

## Introduction

Atherosclerosis is the major cause of morbidity and mortality in the western world. [1] So far fractional-flow-reserve is considered the gold standard for functional lesion interrogation. It has been shown that clinical outcomes of FFR-guided interventions were superior than those of angiography-guided interventions or conservative medical therapy [2]. Recently, a novel, adenosine-free tool for functional assessment of a coronary lesion—quantitative flow ratio (QFR)—was introduced, which is based on quantitative coronary angiography and computational algorithms. [3]

Several targets for noninvasive imaging have been identified for the detection of vulnerable coronary atherosclerotic plaques. [4,5,6] Non-enhanced and contrast-enhanced cardiovascular magnetic resonance imaging (CMR) provide additional information on plaque morphology and biology. For instance noncontrast—enhanced T1-weighted CMR has shown to be feasible for the identification of intraplaque hemorrhage and thrombus. [7,8] Additionally molecular CMR with the use of target-specific contrast agents highlight certain molecules or cells in order to visualize and characterize a pathological processes on the molecular level, which potentially help to better understand molecular events that contribute to coronary plaque formation.

[4,5,9] The albumin-binding probe (gadofosveset-trisodium) investigated in this study is a clinically approved target-specific molecular MR probe and behaves similarly to Evan's blue dye, a marker of endothelial permeability. [4,5,6] It reversibly binds to albumin and was originally designed as a blood pool agent for steady-state angiography, before its use for visualization of endothelial permeability and neovascularization was discovered. [4,5,6]) Leaky endothelial junctions may facilitate migration of macromolecules, such as albumin and leucocytes, into the vessel wall, consequently leading to plaque progression. [4]

Hypoxemia within the growing plaque results in an increase in neoangiogenesis and proliferation of new fragile neovessels with increased endothelial permeability. [4] Rupture-prone

atherosclerotic plaques are most often characterized by the presence of these intraplaque neo-vessels (i.e. neoangiogenesis). [4,5]

These Information from noninvasive CMR may complement information derived from high-resolution invasive plaque analysis such as optical coherence tomography (OCT) for improved characterization of coronary atherosclerosis

However, there is still limited knowledge about morphological imaging-derived parameters to further characterize hemodynamically–significant and prognostic relevant coronary lesions. [10,11] The purpose of this study was to describe and differentiate hemodynamically significant from non-significant coronary lesions as assessed by QFR-angiography, using different types of noninvasive and invasive tools.

## Methods

### Study population

Subjects with symptoms suggestive of coronary artery disease such as stable chest pain and acute coronary syndrome (unstable angina or Non-ST-elevation myocardial infarction; NSTEMI) were prospectively recruited between April 2015 and June 2016 and underwent T1-weigthed non-enhanced cardiovascular magnetic resonance imaging (CMR) and 24 and gadofosveset-enhanced CMR within 24 hours. Subsequently invasive coronary angiography and functional lesion interrogation using quantitative flow ration (QFR) was performed in each patient. Hemodynamically unstable patients (patients with cardiogenic shock, rising cardiac enzymes or malignant cardiac arrhythmias, ST-elevation myocardial infarction; STEMI), pregnant women, patients with a history of coronary stenting, patients with renal insufficiency (creatinine clearance <30 ml/min) and patients, who were not able to give their written consent (i.e. <18 years of age, because of mental disorders), have been excluded from the study. Further exclusion criteria were the presence of common contraindication to cardiac magnetic resonance imaging (i.e. allergy to gadolinium-based contrast agents, claustrophobia, specific metallic–items such as cochlear implants, central nervous system aneurysm clips, pacemakers/defibrillators). Written informed consent was obtained from each subject and the study was approved by the local ethics committee (Ethikkommission, Charité - Universitätsmedizin Berlin) for clinical investigations and performed in accordance with the Declaration of Helsinki.

### Cardiac magnetic resonance imaging

A 3-Tesla MRI scanner (Magnetom Skyra, Siemens Healthcare, Erlangen, Germany) using an 18-channel matrix coil was used in this study. Continuous monitoring of vital signs throughout the entire cardiovascular magnetic resonance imaging (CMR) scan was performed with a 4-lead ECG. Patients with elevated cardiac enzymes, were monitored using a CMR-compatible blood pressure monitor and blood oxygenation sensor. In all patients, CMR imaging was performed twice; natively and 24 hours following the administration of gadofosveset trisodium. Gadofosveset-trisodium (0.03 mmol/kg body weight) was administered intravenously through a catheter in an antecubital vein following the first imaging session. In this study, we used a post-contrast imaging time of around 24 hours to reduce the signal from gadofosveset in the coronary lumen and to get optimal wall to lumen contrast as previously demonstrated by Lobbes et al. [6]

After the acquisition of scout scans to identify the major structures of the heart, a cine 4-chamber-view was used to determine trigger delay and acquisition window, followed by a TI scout to determine the patient-specific inversion time to null signal from blood. For whole heart MR coronary angiography a FLASH (fast low angle shot) sequence (T2-preparation) was used including the following imaging parameters: field of view 340 x 340 mm; acquisition

matrix 256 x 256; reconstruction matrix 512 x 512; acquisition slice thickness 1.3 mm; acquisition slice number 80–100; (physical) spatial inplane resolution 1.3mm x1.3mm, interpolated inplane spatial resolution 0.65 x 0.65 mm; repetition time/echo time 3.5 ms/1.42 ms; flip angle 20 degrees. An inversion recovery (IR) prepared 3-dimensional (3D) T1W turbo FLASH (fast low angle shot) sequence with fat suppression (FatSat) was used for whole heart coronary vessel wall imaging without an interslice gap. Electrocardiogram–triggering and a navigator-gated free breathing technique in coronal orientation was part of each acquisition. To null blood using a region of interest to determine the most accurate value, we adjusted the patient-specific inversion time (range 270 to 300 ms) using the TI scout sequence. The following acquisition parameters were included: inversion time 250±15 ms; field of view 340 x 340 mm; acquisition matrix 256 x 256; reconstruction matrix 512 x 512; acquisition slice thickness 1.3 mm; acquisition slice number 80–100; reconstruction spatial resolution 0.65 x 0.65 x 0.65 mm; repetition time/echo time 4.1 ms/1.3 ms; flip angle 15 degrees. The navigator gating window width was 1.5–2.5 mm. The respiratory navigator gating window width to compensate for breathing motion was set to an acceptance window between 1.5–2.5 mm Depending on the patient's heart rate, the data acquisition window duration time varied from 84 to 120 ms depending. The trigger delay and acquisition window were adjusted according to the phase with minimal motion of the right coronary artery (RCA) as determined by cine MR imaging.

## Invasive catheterization and assessment of the functional significance of a stenosis using quantitative flow ratio (QFR)

Invasive Catherization (IC) was performed according to standard techniques via a transradial or transfemoral approach. All lesions with a stenosis severity greater than 50 percent as estimated by the interventional cardiologist, were used for quantitative flow ratio (QFR) determination. To assess the functional significance of a stenosis, computation of QFR was performed offline using a special software package (QAngio XA 3D prototype, Medis Medical Imaging System, Leiden, the Netherlands). This novel computational approach to derive fractional flow reserve from diagnostic coronary angiography was described in detail previously [3]: Two angiographic projections—at least 25 degrees apart from each other–were chosen and a 3D reconstruction of the interrogated vessel including 3D QCA data was performed. On basis of the following principles, the QFR software computed three QFR pullbacks: 1) the pressure of the coronary arteries stays constant through healthy coronary arteries. 2) the extent of pressure drop is influenced by the stenosis geometry and the flow through the site of luminal narrowing. 3) the stenosis geometry can be described by the relation of the diseased vessel site to the reference vessel site (i.e. healthy vessel wall). 4) Blood flow velocity is preserved distally to the stenosis in relation to the blood flow velocity proximally to the site of stenosis.5) The mass flow rate drops as the diameter of the coronary artery gets smaller distally due to the presence of side branches. QFR measurements were performed by two experienced cardiologists with more than five years of experience in angiographic software programs such as QCA (LCE and YSA).

## OCT image analysis

Optical coherence tomography (OCT) was performed in the culprit vessel of patients where possible (glomerular filtration rate >30 ml/min). OCT images were assessed by two experienced readers (M.J. and L.C.E.), who were at the time of OCT analysis blinded to the cardiovascular magnetic resonace images (CMR), using proprietary software (St. Jude medical). Consensus reading was done by a third investigator (B.B.) in case of disagreement between the two OCT readers. Plaques were evaluated using validated OCT criteria. [12]

## CMR image analysis

We used a dedicated image analysis software for cardiovascular magnetic resonance imaging (CMR) image analysis (Osirix 3.6.1, Geneva, Switzerland). Signal intensity of coronary segments was determined in Gadofosveset-enhanced CMR and the T1-weigthed non-enhanced CMR scan according to a 9-segment model as previously. [5,6] Contrast-to-noise ratio (CNR) was defined as the difference in signal enhancement between the coronary segment and blood divided by the background noise (SI lesion–SI blood /noise). The standard deviation of the signal enhancement in a region of interest ventrally to the patient´s chest was used to obtain the background noise.

## Statistics

We used the SPSS software (IBM SPSS Statistics Version 24) for statistical analysis. An unpaired Student t-test and Mann-Whitney U-Test was applied for comparison of continuous and non-normally distributed variables respectively. Nominal variables were compared among the three groups using chi-square and Fisher exact tests, when appropriate. Continuous variables were reported as mean standard deviation or median with interquartile range (25th and 75th percentiles). Nominal variables were reported as percentage or frequencies, as appropriate. A 2-tailed p-value less than 0.05 was reported as statistically significant.

# Results

Out of 26 patients who underwent T1-weighted non-contrast enhanced cardiovascular magnetic resonance imaging (CMR), Gadofosveset-enhanced CMR and invasive coronary angiography between April 2015 and June 2016, we identified 17 patients who had at least one lesion of 50% stenosis on invasive coronary angiography. These lesions (n = 49 segments) were taken for further functional assessment using quantitative flow ratio (QFR). Coronary segments, which were not correctly visible on invasive coronary angiography due to overlap of other vessels of foreshortening (i.e. wrong projections) and coronary segment which did not fulfill the requirement for QFR determination (i.e. two angiographic projections at least 25 degrees apart from each other), were excluded. As a result, overall n = 14 patients and 28 coronary segments were included in the final analysis. Mean age of all patients was 74.1±10.7. The majority of patient was male (64.3%). Other baseline characteristics such as cardiovascular risk factors, lab values, and medication are shown in Table 1.

## Angiographic and quantitative flow ratio (QFR) data

The total amount of patients in this study having 1-vessel, 2-vessel and 3-vessel coronary artery disease were n = 4, n = 8 and n = 1, respectively. Overall n = 4 patients had one lesion with a vessel-QFR ≤0.8, n = 2 patients had both a lesion with a vessel-QFR≤0.8 and a lesion with a vessel-QFR>0.8 and n = 8 patients had only one lesion with a vessel-QFR>0.8. There were 9 out of 28 stenotic segments which were associated with a vessel-QFR ≤0.8 while 19 out of 28 stenotic segments were associated with a vessel-QFR >0.8. From all hemodynamically significant lesions, n = 4 were located the RCA, n = 5 were located in the LAD and n = 0 were located in the LCX. Hemodynamically non-significant lesions were present in all three coronary vessels (LAD, n = 8; LCX, n = 5; RCA, n = 7). Quantitative flow ratio (QFR) data can be seen in Table 2. Mean length of lesions with a QFR ≤0.8 was 21.7±9.2 mm while mean length of lesions with a QFR >0.8 was 14.1±6.4 mm. Percent area stenosis 71.2±11.5% for hemodynamic significant lesions (QFR<0.8) and 52.3±17.9% for hemodynamically non-significant lesions (p = 0.0017).

**Table 1. Baseline patients´ characteristics and medical treatment upon admission.**

| | All patients (n = 14) |
|---|---|
| Age, y | 74.1±10.7 |
| Male, n (%) | 9 (64.3) |
| **Risk factors** | |
| Hypercholesterolemia, n (%) | 6 (42.9) |
| Hypertension, n (%) | 12 (85.7) |
| Diabetes mellitus, n (%) | 5 (35.7) |
| Smoking, n (%) | 5 (35.7) |
| Family history of CAD, n (%) | 3 (21.4) |
| **Laboratory findings** | |
| Troponin T, ng/ml | 153.1±376.6 |
| CK, UI/l | 167.7±122.0 |
| CK-MB, UI/l | 38.3±40.6 |
| Creatinine, mg/dl | 1.1±0.35 |
| C-reactive protein, mg/dl | 25.5±30.4 |
| Platelets, x $10^9$ | 253.7±59.2 |
| Total cholesterol, mg/dl | 166.5±43.4 |
| Triglyceride, mg/dl | 180.0±133.1 |
| HDL cholesterol, mg/dl | 44.1±15.5 |
| LDL cholesterol, mg/dl | 94.6±35.0 |
| Hemoglobin A1c, % | 6.5±1.2 |
| **Medication** | |
| Aspirin | 10 (71.4) |
| Statin | 5 (35.7) |
| Beta-blocker | 7 (50.0) |
| ACEI and/or ARB | 10 (71.4) |

## OCT analysis

Overall 12 segments were analyzed using optical coherence tomography (OCT) of which 50% (n = 6 segments) were functionally relevant (QFR≤0,8) lesions. 66,7% (4 out of 6) of all vulnerable plaque (i.e. thin-cap fibroatheroma) and 33.3% (2 out of 6) of all non-vulnerable plaque (fibroatheroma) as assessed by OCT were hemodynamically significant lesions. In 66,7% (4/6) of all hemodynamically significant lesions, coronary artery calcification (CAC) was present, whereas in 33.3% (2/6) of all hemodynamically non-significant lesions CAC was found.

## CMR analysis

Segments containing lesions with a QFR ≤0.8 (n = 9) were associated with significantly higher signal enhancement on Gadofosveset-enhanced CMR as compared to segments containing a

**Table 2. Quantitative flow ratio (QFR) data.**

| | Lesion-QFR > 0.8 (n = 19) | Lesion-QFR ≤0.8 (n = 9) | P-Value |
|---|---|---|---|
| Lesion length | 14,1±6,4 | 21,7±9,2 | 0.010 |
| Area stenosis* | 52,3±17.9 | 71,2±11,5 | 0.004 |
| Bending angle | 22.8±15.1 | 34.1±22.8 | 0.090 |

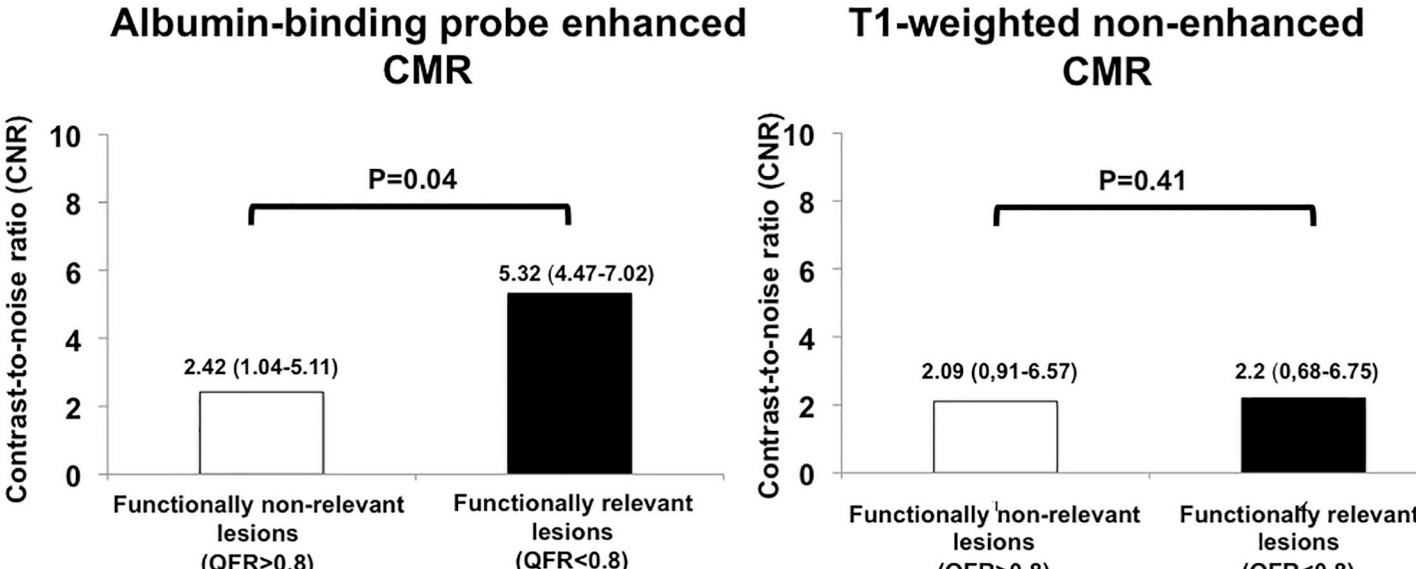

**Fig 1.** Comparison of contrast-to-noise ratios (CNR) between functionally relevant and non-relevant lesions in **A)** gadofosveset-enhanced CMR (i.e. albumin-binding probe enhanced) CMR and **B)** non-contrast-enhanced T1-weighted CMR.

lesion without significant stenosis (lesion-QFR>0.8; n = 19) (5.32 (4.47–7.02) vs. 2.42 (1.04–5.11); p = 0.042). No differences in signal enhancement were seen on native T1-weighted CMR (2.2 (0,68–6.75) vs. 2.09 (0.91–6.57), p = 0.412) (Fig 1). Signal enhancement of segments containing atherosclerotic plaques was 4.0 (2.3–6.4) on gadofosveset-enhanced CMR and 0.7 (-0.9–4.2) on T1-weighted CMR.

## Discussion

Quantitative flow ratio (QFR) was recently introduced as an angiography-based method for deriving fractional flow reserve (FFR) without the use of a pressure wire and demonstrated high diagnostic accuracy in identifying hemodynamically significant coronary stenosis. [3] The main finding of this study is that hemodynamically-significant lesions as assessed by QFR-angiography, were associated with significantly higher signal enhancement on Gadofosveset-enhanced cardiovascular magnetic resonance imaging (CMR) as compared to non-hemodynamically significant lesions (Figs 2 and 3), whereas on native T1-weighted CMR, no differences between the groups were observed. On a pathophysiological level, these findings suggest that functionally relevant lesions have a higher grade of endothelial permeability and / or a higher density of intraplaque neovessels taken into account the findings of prior gadofosveset-enhanced CMR studies, while the presence of intraplaque hemorrhage may not be an appropriate predictor for functionally relevant lesions.

However, it remains to be clarified in a larger study how the albumin leakage sign on molecular CMR can add information to QFR or FFR in cases of borderline obstructive lesions, when decision whether to perform percutaneous coronary intervention (PCI) in invasive coronary angiography or not is unclear.

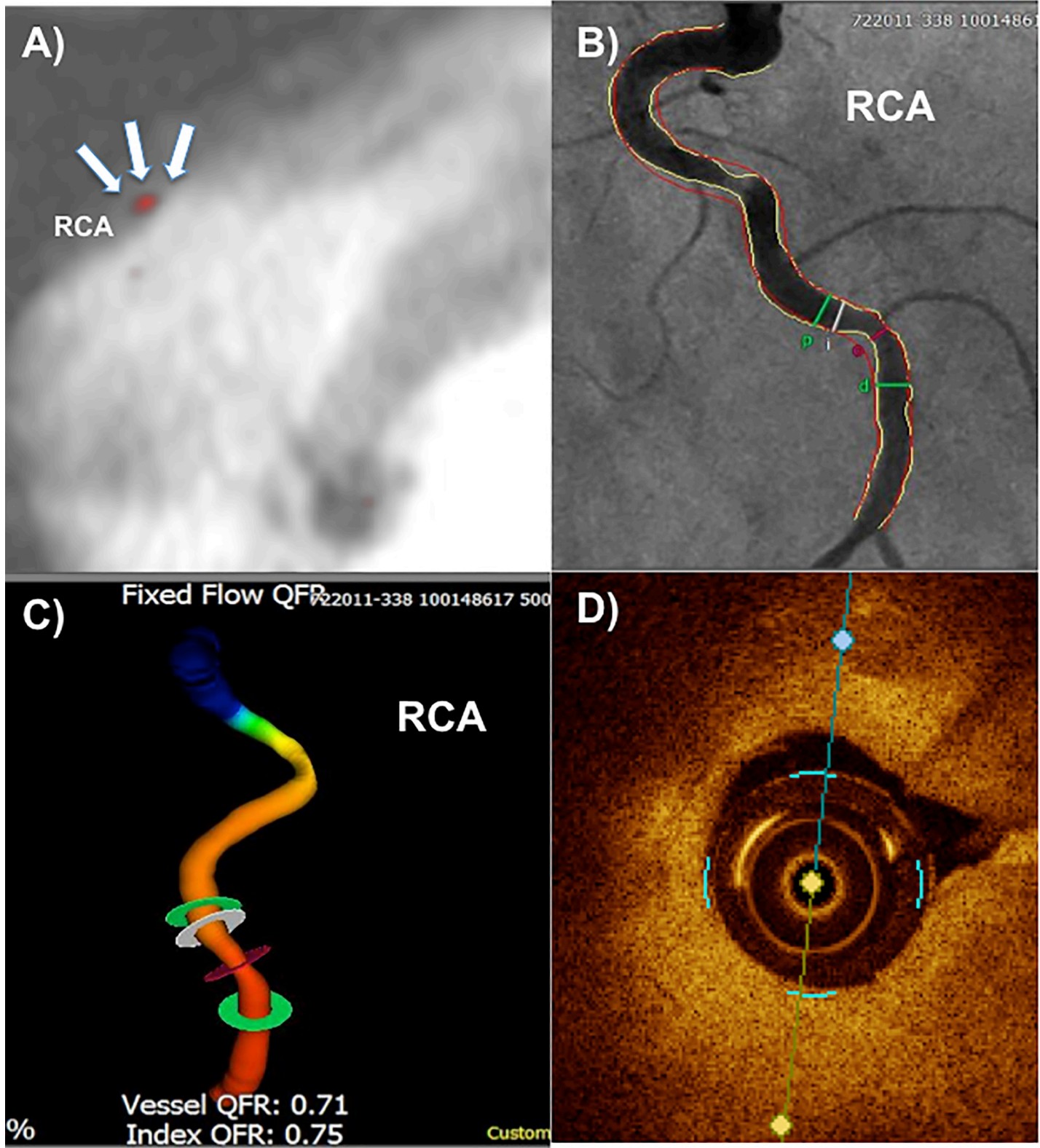

**Fig 2. Sample case of a 64-year-old woman presenting with typical chest pain. A)** Cardiovascular magnetic resonance imaging demonstrated signal enhancement in the mid to distal right coronary artery (RCA). **B)** Invasive catheterization revealed significant stenosis at the site of CMR signal enhancement. **C)** Further analysis using quantitative flow ratio (QFR) graded the stenosis as functionally relevant with a vessel QFR and an index QFR of 0.71 and 0.75 respectively. **D)** Optical coherence tomography, which was performed during invasive catheterization, revealed a fibroatheroma at the site of maximal luminal narrowing.

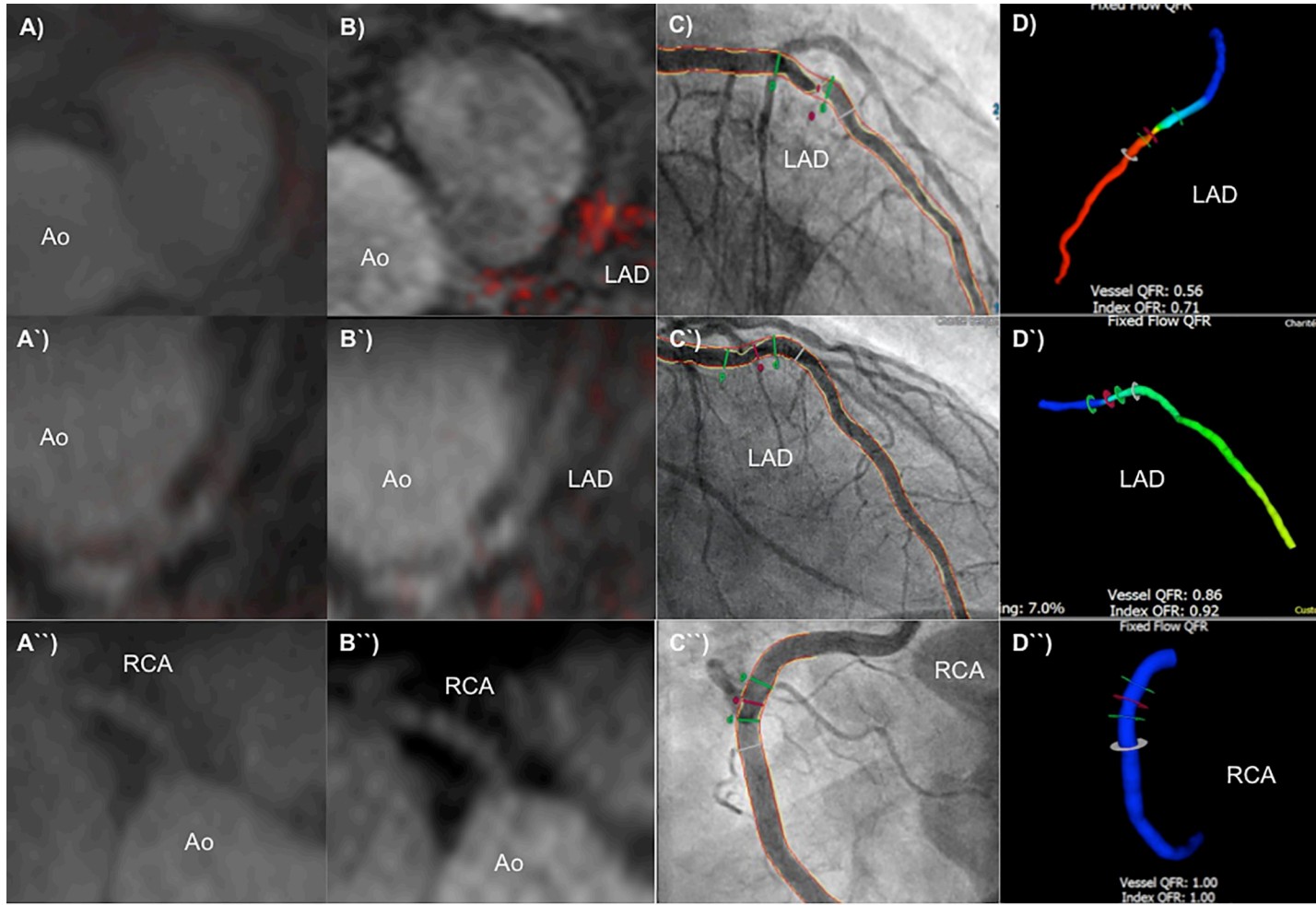

**Fig 3.** Representative images of comprehensive multimodality assessment of coronary lesions with a quantitative flow ratio <0.8 (first row), >0.8 (second row), and non-or minimal diseased vessel wall (third row) using non-enhanced T1-weigthed cardiovascular magnetic resonance imaging (A, A′, A‛), albumin-binding probe–enhanced CMR (B, B′, B′′), invasive catheterization (C, C′, C”) and quantitative flow ratio assessment (D, D′, D”). While T1-weigthed CMR showed no clear difference in signal enhancement between the different lesion categories, gadofosveset-enhanced CMR demonstrated strongest signal enhancmcnt at the site of the hemodynamic-relevant stenosis as compared to the lesion with a QFR below 0,8 and the minimal diseased vessel area; (Ao, aorta; LAD, left anterior descending; RCA, right coronary artery; QFR, quantitative flow ratio).

So far—out of all noninvasive imaging modalities—mainly coronary computed tomography angiography (CCTA) with its relatively high spatial resolution has been able to reliably give quantitative information on plaque characteristics. This allows us to get a clearer understanding of the complex relationship between luminal narrowing and plaque characteristics which defines whether a lesion is functionally–relevant or not. [13] To the best of our knowledge, CMR has not been used to investigate the relationship between the hemodynamic significance of a lesion and its morphologic characteristics but certain CMR sequences have demonstrated to successfully identify high-risk coronary atherosclerotic plaques. [5,14] High-intensity plaques as seen on non-enhanced t1-weighted sequences, for instance, have been considered to be vulnerable lesions. [14] As carotid MRI studies suggested, high-intensity lesions of T1-weighted sequences indicate the presence of methemoglobin in intraplaque hemorrhage which leads to an extensive shortening in T1-relaxation time. [15] Similarly lipid-rich necrotic plaques are associated with strong signal enhancemenet on T1-weighted images, since intraplaque hemorrhage often happens within lipid-rich necrotic cores. [14,15] In a different

approach, MRI in combination with an albumin-binding probe (Gadofosveset) has the potential to identify advanced atherosclerotic plaque based to the presence of endothelial permeability. [4]

Up to now it was believed that the detection of high-risk plaque however—based on either the presence of intraplaque hemorrhage or the presence of endothelial permeability–may not necessarily answer the question whether this lesion was hemodynamically significant, since these plaque were considered as lesions which are not associated with stenosis. In a recent multivariate analysis, however. Driessen et al. were able to demonstrate that certain vulnerable plaque features on CT such as the presence of non-calcified plaque, low-attenuation plaque, positive remodelling, and spotty calcifications, were the only independent predictors of FFR. [16]

In line with this, we actually found in a subset of patients that the majority of vulnerable plaques (i.e. thin-cap fibroatheroma) as assessed using the means of optical coherence tomography (OCT), were hemodynamically-significant lesions with a degree of stenosis of at least 50%.

Our observation that functionally-relevant lesions are associated with high signal enhancement after application of an albumin-binding probe suggests the presence of increased endothelial permeability and endothelial dysfunction at these coronary sites. In fact, this has been shown before in study using IVUS with radiofrequency spectral analysis, where low density plaques with necrotic cores were associated with local endothelial dysfunction. [17,18] Taken together, we hypothesize that local endothelial dysfunction do not allow for sufficient vasodilation during physiological or pharmacological stress, consequently resulting in ischemia, as opposed to coronary lesions without necrotic cores that maintain a vasodilatory capacity.

Our subanalysis using optical coherence tomography (OCT) revealed that coronary artery calcification (CAC) was more common in segments containing lesions with a QFR <0.80. The reason for this observation may be that plaques without CAC were simply smaller and therefore less stenotic than plaques associated with CAC which can be regarded as a marker for plaque size. However the assumption that CAC is associated with a lower FFR/QFR is still controversial. [10,11] For instance in a recent CCTA study, Baraskan et al. concluded that CAC was not an independent predictor of hemodynamically significant stenosis, even though it closely correlated with total and non-calcified plaque volume. [10]

This study has some limitations. The current study consisted only of a relatively small number of patients. Functional lesion interrogation was only performed using QFR-angiography and not using the gold standard adenosin-induced FFR.

Finally, GE-CMR is still associated with a relatively long scan time for the assessment of the coronary enhancement in vivo, which currently limits the applicability of this technique in a wide clinical setting. Using more advanced motion correction techniques in combination with undersampled image reconstruction (e.g. compressed sensing), this limitation may be solved in the future. [19]

## Conclusion

The findings of this pilot study suggest that signal enhancement on albumin-binding probe-enhanced CMR but not on T1-weighted CMR is associated with hemodynamically relevant coronary lesions. Larger studies are needed to validate our findings.

## Author Contributions

**Conceptualization:** Andreas Schuster, Michel Noutsias, Rene M. Botnar, Marcus R. Makowski.

**Data curation:** Leif-Christopher Engel, Kevin Gigengack.

**Formal analysis:** Youssef S. Abdelwahed, Milosz Jaguszewski, Kevin Gigengack.

**Funding acquisition:** Ulf Landmesser, Carsten Skurk, Marcus R. Makowski, Boris Bigalke.

**Investigation:** Youssef S. Abdelwahed, Carsten Skurk.

**Methodology:** Leif-Christopher Engel, Ulf Landmesser, Thomas-Heinrich Wurster, Costantina Manes, Andreas Schuster, Rene M. Botnar, Marcus R. Makowski, Boris Bigalke.

**Project administration:** Leif-Christopher Engel, Ulf Landmesser, Thomas-Heinrich Wurster, Costantina Manes.

**Software:** Leif-Christopher Engel, Milosz Jaguszewski, Costantina Manes, Marcus R. Makowski.

**Supervision:** Bernd Hamm, Rene M. Botnar, Marcus R. Makowski, Boris Bigalke.

**Visualization:** Michel Noutsias.

**Writing – original draft:** Leif-Christopher Engel, Marcus R. Makowski, Boris Bigalke.

**Writing – review & editing:** Ulf Landmesser, Youssef S. Abdelwahed, Kevin Gigengack, Thomas-Heinrich Wurster, Carsten Skurk, Andreas Schuster, Bernd Hamm, Rene M. Botnar.

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
