## [Decision Letter · Decision Letter 0]

11 Sep 2019

PONE-D-19-21636

Comprehensive Multimodality Characterization of Hemodynamically Significant and Non-Significant Coronary Lesion using Invasive and Noninvasive Measures

PLOS ONE

Dear Dr. Engel,

Thank you for submitting your manuscript to PLOS ONE. After careful consideration, we feel that it has merit but does not fully meet PLOS ONE’s publication criteria as it currently stands. Therefore, we invite you to submit a revised version of the manuscript that addresses the points raised during the review process.

We would appreciate receiving your revised manuscript by Oct 26 2019 11:59PM. To enhance the reproducibility of your results, we recommend that if applicable you deposit your laboratory protocols in protocols.io, where a protocol can be assigned its own identifier (DOI) such that it can be cited independently in the future. For instructions see: http://journals.plos.org/plosone/s/submission-guidelines#loc-laboratory-protocols

We look forward to receiving your revised manuscript.

Kind regards,

Wolfgang Rudolf Bauer, M.D., Ph.D.

Academic Editor

PLOS ONE

Journal Requirements:

2. We noted in your submission details that a portion of your manuscript may have been presented or published elsewhere. "Some patients included in this study were also part of the patient cohort of the study "Novel Approach for In Vivo Detection of Vulnerable Coronary Plaques Using Molecular 3-T CMR Imaging With an Albumin-Binding Probe" where - unlike the current study - CMR results were correlated to optical coherence tomography and which was published in JACC cardiovasc imaging in Feb 2019. "  Please clarify whether this [c publication was peer-reviewed and formally published. If this work was previously peer-reviewed and published, in the cover letter please provide the reason that this work does not constitute dual publication and should be included in the current manuscript.

4. Please ensure that you refer to Figures 1 and 2 in your text as, if accepted, production will need this reference to link the reader to the figure.

5. We note you have included a table to which you do not refer in the text of your manuscript. Please ensure that you refer to Table 2 in your text; if accepted, production will need this reference to link the reader to the Table.

Reviewers' comments:

Reviewer's Responses to Questions

**Comments to the Author**

1. Is the manuscript technically sound, and do the data support the conclusions?

Reviewer #1: Partly

2. Has the statistical analysis been performed appropriately and rigorously? 

Reviewer #1: I Don't Know

3. Have the authors made all data underlying the findings in their manuscript fully available?

Reviewer #1: No

4. Is the manuscript presented in an intelligible fashion and written in standard English?

Reviewer #1: Yes

5. Review Comments to the Author

Reviewer #1: Authors presents results on the important topic of the analysis of atherosclerotic malformation in coronaries. The methodology based on validation of the result of contrast enhanced MRI using the albumin-bind extra-vascular contrast media by the measurements of QFR is technically sound. Yet the approach used by the authors in terms of MRI techniques does not look fully adequate to the problem.

The physical spatial resolution of the GRE-based sequence used by authors is above is 1.3 mm /pixel. Additionally , authors use a non-selective single inversion recovery (for 3D sequence)with fat suppression and breath-free navigation for the full-heart vessel -wall imaging. This approach is know for introducing significant quantitative bias in results based on signal intensity analysis due to incomplete suppression of fat signal and related problems of identification vessel wall and especially adequate segmenting of the wall malformation. For this reason, number of researches introduced double and even quadruple IR methods for separation of signal from blood, tissue , fat and contrast media in order to make quantitative analysis more reliable. In this work when relatively simple approach is used it would be important to demonstrate clearly the adequate quality of CMR images for two considered cases ( QFR>0.8 and QFR<0.8) especially the quality of blood and fat suppression and to show the example of identified lesion segments images along with non-lesion vessel wall. This would support the capability of methodology to provide conclusive and diagnostically relevant results. Currently paper contains only single CMR image with relatively low spatial resolution and quality of blood suppression. The methodology of data processing on this point should be uncovered in more details. The same is for the quality of the respiratory gating which is known to be drastic problem for the CMR and demands at least demonstration of reliability and comparison with breath-hold technique to be able to estimate the severity of introduced artifacts.

Paper has relative broad Discussion in which however is difficult to identify own result of author from the results from literature. On the other hand Introduction to very concise and no adequate overview of the existing works on coronary vessel wall MRI and existing problems of analysis and quantification is given whereas this topic is covered by numerous publication in the specialized MRI journals ( Magnetic Resonance in Medicine in particular). Most part of the discussion would be reasonable to transfer to the Introduction and update the later one with more references to the key works in vessel wall MR-imaging and especially its technical challenges.

The Conclusion is short, but fussy and does not really match to the formulated objectives of the paper. With limited amount of scanned patients and being a pilot study the focus should be given to the workflow and technical details of the approach to prove the feasibility of methods and evaluate reasoning for the further patient recruitment or necessity to work on technical side.

Some specific remarks

Line 63 – 65 abbriviation CMR, OCT, CAD are not defined.

Line 81 - What does it mean “more advanced” . “Conclusion” does not really match to the goal formulated in “Objective”. Which experimentally measured parameters _parameters_ and values allows or does not allow do make conclusions regarding significance of lesions ?

Line 104. Why the role of CMR measurements is not mentioned in Introduction ? Why it is mentioned in Abstract but no details on the capabilities of MRI for atherosclerosis detection are given here and very little overview of the existing methods of atherosclerotic plaques detection and analysis ( especially contrast enhanced vessel wall imaging). The same is about role MR-angiography and MR-based flow quantification – no or very poor literature overview for the Introduction for paper where CMR supposed to provide the key results.

Line 166 „T2-prepared“ ?

Line 170 : 340mm FOV at matrix 256x256 provides resolution (physical )1.3mm. 0.65mm is interpolated resolution.

169: What was the interslice gap ? Was it 2D ( slice selective RF-pulses) or 3D-encoded sequence ?

170: 3.5ms TR time is quite short and hardware demanding. Was it a standard vendor -provided protocol ( e.g TWIST or VIBE ) or any inhouse-developed sequence ?

182: What means “navigator gating window” in “mm”. Usually “window” for the navigator is time available for measurements the same as cardiac trigger window. Proper adjustment of navigator position is essential part of such measurements – some data demonstrating the navigation quality ( as shown by scanner) and dependence of image quality on the position of navigator “pencil” would be very helpful for the illustration of method robustness and role of motion artifacts in the results.

Line 211: The scheme illustrating the segmenting of coronary is key point of data processing. It would be helpful to demonstrate 1 or 2 examples with significant and non-significant lesion for understanding of the methodology and it reliability.

278: The statistics of quantitative results does not look very reliable for CMR. The standard deviation of enhancement is close or even exceeds ( for native T1) the mean value. What is explanation of such a major variation. How reliable is this data and are they reproducible within single patient ?

6. PLOS authors have the option to publish the peer review history of their article (what does this mean?). If published, this will include your full peer review and any attached files.

Reviewer #1: No

---

## [Author Response · Author response to Decision Letter 0]

25 Dec 2019

Dear Prof. Bauer, M.D., Ph.D.,

Here we are submitting the revision of the above mentioned manuscript to “PLOS ONE”. 

We have addressed all remaining concerns and suggestions that you and the reviewers have provided and hope that these changes would increase the priority for publication in PLOS ONE.

As mentioned in the initial submission, the current study consists of 11 patients, who were also part of our prior study "Novel Approach for In Vivo Detection of Vulnerable Coronary Plaques Using Molecular 3-T CMR Imaging With an Albumin-Binding Probe" where - unlike the current study - CMR results were solely correlated to optical coherence tomography and which was published in JACC cardiovasc imaging in Feb 2019. On the contrast, this current study correlates findings of both Gadofosveset-enhanced CMR and non-enhanced T1-weighted CMR to a novel angiographic-based method to assess the hemodynamic relevance of a coronary lesion (i.e. quantitative flow ratio; QFR). Therefore we believe that this study provide a does not constitute dual publication.

Again, we want to express our appreciation for the time and effort you and each of the reviewers have dedicated to provide insightful feedback on ways to strengthen our manuscript.

To facilitate your review of our revisions, the following is a point-by-point response to the reviewers’ questions and comments delivered in your letter dated on September 29 th, 2019.

We look forward to hearing from you regarding our revision. We would be glad to respond to any further questions and comments that you may have. 

Signed respectfully on behalf of all authors,

Leif-Christopher Engel, MD &

Marcus Makowski, MD &

Boris Bigalke, MD, MBA

Reviewer: 1

Comment1 / reviewer#1

The authors have done a great effort and the manuscript has improved notably but unfortunately there are still issues that need to be corrected. To help finishing this task in the hope that the authors provide a high quality report that the journal may consider I provide a detailed recipe of corrections:

Response to comment 1 / reviewer #1

We are grateful for the effort and time the reviewer spent on improving our manuscript. In the following, we have addressed all of the remaining concerns.

5. Review Comments to the Author

comment 2 / reviewer #1

Reviewer #1: Authors presents results on the important topic of the analysis of atherosclerotic malformation in coronaries. The methodology based on validation of the result of contrast enhanced MRI using the albumin-bind extra-vascular contrast media by the measurements of QFR is technically sound. 

• Response to comment 1 / reviewer #1

We are grateful for the effort and time the reviewer spent on improving our manuscript and appreciate this evaluation. In the following, we have addressed all of the remaining concerns.

comment 3 / reviewer #1

Yet the approach used by the authors in terms of MRI techniques does not look fully adequate to the problem. The physical spatial resolution of the GRE-based sequence used by authors is above is 1.3 mm /pixel. Additionally, authors use a non-selective single inversion recovery (for 3D sequence) with fat suppression and breath-free navigation for the full-heart vessel - wall imaging. This approach is know for introducing significant quantitative bias in results based on signal intensity analysis due to incomplete suppression of fat signal and related problems of identification vessel wall and especially adequate segmenting of the wall malformation. For this reason, number of researches introduced double and even quadruple IR methods for separation of signal from blood, tissue, fat and contrast media in order to make quantitative analysis more reliable. 

• Response to comment 3 / reviewer #1

We agree with the reviewer, that double and even quadruple IR methods are more advance methods for this type of acquisition. In the patient collective we investigated the non-selective single inversion recovery (for 3D sequence) with fat suppression and breath-free navigation for the full-heart vessel - wall imaging however was a reliable approach and we did not observe significant issues with fat suppression in the mediastinum in this study. We however fully agree that for future studies a double and quadruple IR could improve the image quality further. 

comment 4 / reviewer #1

In this work when relatively simple approach is used it would be important to demonstrate clearly the adequate quality of CMR images for two considered cases ( QFR>0.8 and QFR<0.8) especially the quality of blood and fat suppression and to show the example of identified lesion segments images along with non-lesion vessel wall. This would support the capability of methodology to provide conclusive and diagnostically relevant results. 

• Response to comment 4 / reviewer #1

We have included sample pictures for multimodality assessement for overall three different lesions types (QFR>0.8, QFR<0.8 and minimal diseased vessel wall), emphasizing the potential of gadofosveset-enhanced CMR to identify hemodynamic-relevant coronary lesions.

comment 5 / reviewer #1

Currently paper contains only single CMR image with relatively low spatial resolution and quality of blood suppression. The methodology of data processing on this point should be uncovered in more details. The same is for the quality of the respiratory gating which is known to be drastic problem for the CMR and demands at least demonstration of reliability and comparison with breath-hold technique to be able to estimate the severity of introduced artifacts.

• Response to comment 5 / reviewer #1

We thank the reviewer for this suggestion. As written before, we have included more samples of multimodality assessment of lesions of different hemodynamic relevance. We acknowledge the limitations of the current CMR protocol. However, a comparable CMR protocol was successfully used in several prior CMR studies 

comment 6 / reviewer #1

• Paper has relative broad Discussion in which however is difficult to identify own result of author from the results from literature. On the other hand Introduction to very concise and no adequate overview of the existing works on coronary vessel wall MRI and existing problems of analysis and quantification is given whereas this topic is covered by numerous publication in the specialized MRI journals (Magnetic Resonance in Medicine in particular). Most part of the discussion would be reasonable to transfer to the Introduction and update the later one with more references to the key works in vessel wall MR-imaging and especially its technical challenges.

• Response to comment 6 / reviewer #1

We appreciate the comment of the reviewer and have accordingly shortened the discussion. The introduction was extended with information on CMR plaque evaluation.

comment 7 / reviewer #1

The Conclusion is short, but fussy and does not really match to the formulated objectives of the paper. With limited amount of scanned patients and being a pilot study the focus should be given to the workflow and technical details of the approach to prove the feasibility of methods and evaluate reasoning for the further patient recruitment or necessity to work on technical side.

• Response to comment 7 / reviewer #1

We have changed the conclusion section accordingly, better matching now to the objectives of the current study.

comment 8 / reviewer #1

Line 63 – 65 abbriviation CMR, OCT, CAD are not defined.

• Response to comment 8 / reviewer #1

The above mentioned abbreviation are now explained (i.e. CMR, cardiovascular magnetic resonance imaging; OCT, optical coherence tomography; CAD, coronary artery disease)

comment 9 / reviewer #1

Line 81 - What does it mean “more advanced” 

• Response to comment 9 / reviewer #1

More advanced in this context means that plaque associated the a QFR<0,8 may be associated with a higher cardiovascular risk since OCT revealed a higher proportion of vulnerable plaques in these lesion

comment 10 / reviewer #1

“Conclusion” does not really match to the goal formulated in “Objective”. 

• Response to comment 10 / reviewer #1

We have changed the conclusion part so that it matches the objectives section

comment 11 / reviewer #1

Which experimentally measured parameters _parameters_ and values allows or does not allow do make conclusions regarding significance of lesions ?

• Response to comment 11 / reviewer #1

Obviously lesion length, area stenosis and bending angle were parameter that significantly different between hemodynamically significant and non-significant lesions. Other – mainly qualitative parameters such as the presence of coronary artery calcification within the lesion – did not differ between lesions with a QFR greater or smaller than 0,8.

comment 12 / reviewer #1

Line 104. Why the role of CMR measurements is not mentioned in Introduction? Why it is mentioned in Abstract but no details on the capabilities of MRI for atherosclerosis detection are given here and very little overview of the existing methods of atherosclerotic plaques detection and analysis ( especially contrast enhanced vessel wall imaging). The same is about role MR-angiography and MR-based flow quantification – no or very poor literature overview for the Introduction for paper where CMR supposed to provide the key results.

• Response to comment 12 / reviewer #1

Thank you for this comment. We have now edited and extended the introduction to include more background information on CMR and its role on plaque evaluation and vessel wall imaging.

comment 13 / reviewer #1

Line 166 „T2-prepared“ ?

• Response to comment 13 / reviewer #1

We apologize for this typo. „T2-prepared“ was changed to „T2- preparation “

comment 14 / reviewer #1

Line 170 : 340mm FOV at matrix 256x256 provides resolution (physical) 1.3mm. 0.65mm is interpolated resolution.

• Response to comment 14 / reviewer #1

We apologize for this inaccuracy and are now clearly stating the resolutions achieved: 

“…following imaging parameters: field of view 340 x 340 mm; acquisition matrix 256 x 256; reconstruction matrix 512 x 512; acquisition slice thickness 1.3 mm; acquisition slice number 80 - 100; (physical) spatial inplane resolution 1.3mm x1.3mm, interpolated inplane spatial resolution 0.65 x 0.65 mm.

comment 15 / reviewer #1

169: What was the interslice gap ? Was it 2D ( slice selective RF-pulses) or 3D-encoded sequence ?

• Response to comment X / reviewer #1

It was a 3D-encoded sequence sequence without an interslice gap. We are now clarifying this in the manuscript.

comment 16 / reviewer #1

170: 3.5ms TR time is quite short and hardware demanding. Was it a standard vendor -provided protocol ( e.g TWIST or VIBE ) or any inhouse-developed sequence ?

• Response to comment 16 / reviewer #1

The sequence was based on a standard vendor-provided protocol.

comment 17 / reviewer #1

182: What means “navigator gating window” in “mm”. Usually “window” for the navigator is time available for measurements the same as cardiac trigger window. Proper adjustment of navigator position is essential part of such measurements – some data demonstrating the navigation quality ( as shown by scanner) and dependence of image quality on the position of navigator “pencil” would be very helpful for the illustration of method robustness and role of motion artifacts in the results.

• Response to comment 17 / reviewer #1

We apologize for being not clearer regarding the naviator. We are now claryfing this in the manuscript: The respiratory navigator gating window width to compensate for breathing motion was set to an acceptance window between 1.5-2.5 mm.

comment 18 / reviewer #1

Line 211: The scheme illustrating the segmenting of coronary is key point of data processing. It would be helpful to demonstrate 1 or 2 examples with significant and non-significant lesion for understanding of the methodology and it reliability.

• Response to comment 18 / reviewer #1

We thank the reviewer fort his suggestion and have included a corresponding figure.

comment 19 / reviewer #1

278: The statistics of quantitative results does not look very reliable for CMR. The standard deviation of enhancement is close or even exceeds ( for native T1) the mean value. What is explanation of such a major variation. How reliable is this data and are they reproducible within single patient ?

• Response to comment 19 / reviewer #1

We thank the reviewer fort his suggestion. The data is not normally distributed. Therefore we have used a „mann whitney-U-Test“ comparing median values.

---

## [Editor Report · Decision Letter 1]

13 Jan 2020

Comprehensive Multimodality Characterization of Hemodynamically Significant and Non-Significant Coronary Lesions using Invasive and Noninvasive Measures

PONE-D-19-21636R1

Dear Dr. Engel,

We are pleased to inform you that your manuscript has been judged scientifically suitable for publication and will be formally accepted for publication once it complies with all outstanding technical requirements.

With kind regards,

Wolfgang Rudolf Bauer, M.D., Ph.D.

Academic Editor

PLOS ONE

---

## [Editor Report · Acceptance letter]

24 Jan 2020

PONE-D-19-21636R1 

Comprehensive Multimodality Characterization of Hemodynamically Significant and Non-Significant Coronary Lesions using Invasive and Noninvasive Measures 

Dear Dr. Engel:

I am pleased to inform you that your manuscript has been deemed suitable for publication in PLOS ONE. Congratulations! Your manuscript is now with our production department. 

With kind regards,

on behalf of

Prof. Wolfgang Rudolf Bauer 

Academic Editor

PLOS ONE